

# Proactive detection of anomalous behavior in Ethereum accounts using XAI-enabled ensemble stacking with Bayesian optimization

Vasavi Chithanuru and Mangayarkarasi Ramaiah

School of Computer Science Engineering and Information Systems, Vellore Institute of Technology, Vellore, Tamilnadu, India

## ABSTRACT

The decentralized, open-source architecture of blockchain technology, exemplified by the Ethereum platform, has transformed online transactions by enabling secure and transparent exchanges. However, this architecture also exposes the network to various security threats that cyber attackers can exploit. Detecting suspicious behaviors in account on the Ethereum blockchain can help mitigate attacks, including phishing, Ponzi schemes, eclipse attacks, Sybil attacks, and distributed denial of service (DDoS) incidents. The proposed system introduces an ensemble stacking model combining Random Forest (RF), eXtreme Gradient Boosting (XGBoost), and a neural network (NN) to detect potential threats within the Ethereum platform. The ensemble model is fine-tuned using Bayesian optimization to enhance predictive accuracy, while explainable artificial intelligence (XAI) tools—SHAP, LIME, and ELI5—provide interpretable feature insights, improving transparency in model predictions. The dataset used comprises 9,841 Ethereum transactions across 52 initial fields (reduced to 17 relevant features), encompassing both legitimate and fraudulent records. The experimental findings demonstrate that the proposed model achieves a superior accuracy of 99.6%, outperforming that of other cutting-edge methods. These findings demonstrate that the XAI-enabled ensemble stacking model offers a highly effective, interpretable solution for blockchain security, strengthening trust and reliability within the Ethereum ecosystem.

## INTRODUCTION

Blockchain technology, valued for its decentralized, secure, and unalterable nature, has become pivotal in strengthening security across sectors such as governance, healthcare, financial systems, and urban development (*Ramaiah et al., 2022*; *Padma & Ramaiah, 2024a*). Ethereum stands out as a leading blockchain platform, utilizing smart contracts and its native cryptocurrency for network operations. Ethereum relies on distinct account identifiers to facilitate transactions, including externally owned accounts (EOAs) secured by private keys and contract accounts governed by code. To maintain data integrity, it

Corresponding author
Mangayarkarasi Ramaiah,
rmangayarkarasi@vit.ac.in

utilizes features such as the nonce, contract code hash, and storage root, with the Keccak-256 hashing algorithm ensuring the authenticity of smart contracts (*Siddique & Fatima, 2022*). The Ethereum storage root, based on the Merkle Patricia Trie, serves as a unique account identifier, ensuring efficient data encoding and verification (*Wang et al., 2021*) and (*Chen et al., 2020a*).

As of August 2024, Ethereum has processed over 2.4 billion transactions (*Etherscan, 2024*), making its decentralized ledger a target for cyber threats. The anonymity within Ethereum enables malicious activities such as money laundering and illicit goods sales. Vulnerabilities within the Ethereum network, including P2P limitations and smart contract exploits, expose it to attacks like phishing (*Chen et al., 2020c*; *Kabla et al., 2022*), Ponzi schemes (*Chen et al., 2019*), the DAO attack, and 51% attack (*Scicchitano et al., 2020*), as well as the exploitation of malicious contracts (*Wen et al., 2021*), accounts (*Kumar et al., 2020*), eclipse attacks (*Xu et al., 2020*), and abnormal smart contracts (*Liu et al., 2022*).

Ethereum's blockchain technology is renowned for its transparency, resistance to tampering, and immutable nature, offering robust security capabilities.Despite these strengths, malicious actors have successfully identified and exploited vulnerabilities within elements such as smart contracts (*Kushwaha et al., 2022*; *Padma & Mangayarkarasi, 2022*), the Solidity programming language (*Kaleem, Mavridou & Laszka, 2020*) and the Ethereum architecture (*Chen et al., 2020b*). However, a range of techniques and tools (*Ramaiah et al., 2022*) has emerged to actively monitor and identify malicious activities within Ethereum networks. Ongoing efforts focus on developing solutions to enhance the resilience of the Ethereum ecosystem against potential threats. Despite the existence of vulnerabilities within the Ethereum blockchain (*Farrugia, Ellul & Azzopardi, 2020*), continuous advancements aim to strengthen security measures and minimize the occurrences of fraudulent activities.

Researchers are actively engaged in addressing security concerns within Ethereum's system while ensuring optimal performance. The application of artificial intelligence (AI) technology has tremendous potential for early identification of security vulnerabilities (*Padma & Ramaiah, 2024b*). As highlighted by *Kumar et al. (2020)*, machine learning (ML) models were employed to distinguish between malicious and legitimate Ethereum addresses, with a focus on externally owned accounts (EOAs) and smart contract accounts. By extracting key transaction features, models like eXtreme Gradient Boosting (XGBoost), Random Forest (RF), and K-nearest neighbors (KNN) demonstrated strong performance in enhancing Ethereum security, particularly for smart contract accounts. Also, Ponzi schemes represent significant financial threats, particularly in vulnerable communities like Nigeria. A study by *Onu et al. (2023)* applies AI with ML models, including RF, NN, and KNN, to detect these schemes on Ethereum by analyzing transaction patterns. AI-driven models offer a promising approach to minimizing security breaches effectively. The convergence of Ethereum and AI technologies aims to improve the platform's capabilities, bolster its security measures, and enhance the overall user experience, delivering advantages to both individual users and businesses relying on the Ethereum network. *Aziz et al. (2023)* explores strategies to optimize Ethereum transactions

and successfully addresses deviations. Despite notable strides in AI-driven detection of malicious activities within the Ethereum platform, substantial enhancements remain necessary. Challenges ahead of AI enabled malicious activities detection system are suitable feature engineering (*Rahamathulla & Ramaiah, 2024*) enough samples (*Ramaiah et al., 2024*) and suitable hyperparameter (*Zhao et al., 2023*) searching technique.

AI models produce outcomes, but we need clearer insights into how they reach those conclusions. That's where explainable artificial intelligence (XAI) comes into build trust in AI decisions by shedding light on their reasoning and holding them accountable. XAI pertains to the application of AI technology in a manner that allows human experts to comprehend the outcomes of the solution. This differs from the concept of a "black box" in machine learning, when even the developers are unable to explicate the reasoning behind a single AI choice. XAI serves as an embodiment of the social right to an explanation. The evolving field of XAI introduces various methods aimed at transforming the opaque nature of models based on ML or deep learning (DL), thereby generating explanations that are intelligible to humans. With the remarkable progress in ML and DL, researchers in the fields of AI and ML are increasingly prioritizing the development and application of XAI. Researchers have developed various tools to demystify black-box models, such as Local Interpretable Model-Agnostic Explanations (LIME), Shapley Additive Explanations (SHAP), Explain Like I am a 5-year old (ELI5), and InterpretML, among others (*Buyuktepe et al., 2023*).

Due to the enhanced merits, XAI finds application in healthcare (*Hauser et al., 2022*), cyber-attack detection (*Kalutharage et al., 2023*). These techniques are also used in various fields like fraud detection (*Biswas et al., 2023*; *Zhou et al., 2023*), cyber security (*Rjoub et al., 2023*), smart cities (*Javed et al., 2023*), Internet of Things (*Kök et al., 2023*), and intelligent connected vehicles (*Nwakanma et al., 2023*).

The growing adoption of Ethereum has also heightened its susceptibility to various cyber threats. While previous studies using ML models for anomaly detection have demonstrated promise, they often face challenges with interpretability and scalability when handling large transaction volumes. Moreover, current anomaly detection approaches face challenges in balancing precision and computational efficiency, frequently operating as "black-box" models that offer limited transparency.

Hence, the study presented in this article aims to develop comprehensive XAI-enabled models that integrate the strengths of three ML models—RF, XGBoost, and NN—into an ensemble stacking model. To improve prediction accuracy, a Bayesian optimization technique has been proposed to optimize the control parameters of these ML models. To address the sample imbalance issue, an appropriate oversampling technique has been applied. Above all, the experiment leverages the advantages of XAI tools, including SHAP, LIME, and ELI5, for feature interpretation. A well-designed ensemble ML model has been constructed effectively detect fraudulent transactions, aiming to strengthen the Ethereum network against possible security vulnerabilities. Through the examination of model transparency and feature influence, this research seeks to enhance fraud detection on Ethereum while improving the interpretability and trustworthiness of the proposed model. This aligns with the growing demand for explainable AI in cybersecurity applications.

## Major contributions in this article

- To mitigate the data sample imbalance, oversampling technique is deployed.
- Bayesian optimization technique is implemented to decide the control parameters of the ML models.
- Implementation of XAI techniques (SHAP, LIME, and ELI5) to interpret feature importance, adding an explainability layer to the model and improving decision transparency.
- An ensemble stacking model that combines XGBoost, RF, and NN to provide robust fraud detection capabilities.
- Extensive experimental validation demonstrating a model accuracy of 99.6%, benchmarked against existing state-of-the-art solutions.

## Paper organization

The manuscript is organized as follows: "Literature Review" reviews related literature and highlights its limitations. "Materials and Methods" outlines the materials and methods employed in the proposed framework. "Results and Discussion" presents the framework's performance and compares it with existing methods. Finally, "Conclusion" concludes the study and offers suggestions for future research directions.

## LITERATURE REVIEW

Using smart contracts, Ethereum is a blockchain-based platform that lets one create distributed apps and handle bitcoin transactions. The platform's "pseudo-anonymous" structure allows users to maintain multiple accounts under distinct cryptographic identities, complicating the detection and attribution of fraudulent activity. High-profile incidents, such as the Ethereum DAO attack—where hackers exploited a smart contract vulnerability to steal cryptocurrency—underscore these challenges. Fraudulent activities across multiple identities are significant security risks, highlighting the need for effective monitoring to protect legitimate users. *Kumar et al. (2020)* tackles these challenges with training supervised ML models to identify malicious and legitimate addresses on the Ethereum network, focusing on EOAs and smart contract accounts. Malicious and validated non-malicious addresses were acquired from various sources, followed by extensive data preprocessing to differentiate and categorize EOAs and contract accounts. Important features were retrieved from transaction data to train models including RF, Decision Tree, XGBoost, and KNN for both account types. The approach XGBoost achieved high accuracy rates 96.82% for smart contract accounts, demonstrating the efficacy of ML in enhancing security within the Ethereum ecosystem.

In their study, *Farrugia, Ellul & Azzopardi (2020)* introduced an innovative approach using the XGBoost model to detect illicit accounts on the Ethereum. The researchers conducted an in-depth analysis of account details, such as addresses and transaction histories, to uncover key insights for detecting suspicious activities. By employing a careful feature selection process, they pinpointed the most influential factors affecting the model's

predictions. The study's findings emphasized the significance of features like total ether balance, transaction duration, and minimum transaction value in enhancing the model's performance. This novel approach has the potential to significantly reduce illegal activities on the Ethereum network, including phishing, bribery, and money laundering.

To optimize classification performance, *Zhou, Yan & Zhang (2022)* observed sample distribution using t-SNE and K-means clustering techniques.To detect the fraudulent transaction on Ethereum platform, a CatBoost based ML model upon the data distribution analysis by T-SNE and K-Means has been presented by following this analysis, they crafted a ML model based on CatBoost to mitigate the impact of fraudulent transactions within Ethereum accounts and used SHAP to find the feature importance.

Ethereum, as a digital currency, faces growing fraudulent activities like money laundering and phishing, threatening transaction security. This research advocates the application of the light gradient boosting machine (LGBM) algorithm for the identification of fraudulent Ethereum transactions, juxtaposing it with RF and multi-layer perceptron (MLP) models. A comparison of bagging models shows that LGBM and XGBoost achieve the highest accuracies, with LGBM slightly outperforming XGBoost at 98.60%. By tuning LGBM's hyperparameters, an accuracy of 99.03% is achieved (*Aziz et al., 2022*). *Ibrahim, Elian & Ababneh (2021)* proposed a fraud detection model for Ethereum using decision tree, RF, and KNN algorithms. They selected six key features from a Kaggle dataset, and RF outperformed the others in processing time and F-measure. This highlights RF effectiveness in detecting Ethereum fraud.

*Feichtner & Gruber (2020)* introduced an XAI-enabled CNN model to connect app descriptions with requested permissions, using LIME heatmaps to visualize word significance. *Hsupeng et al. (2022)* developed an explainable flow-data categorization system to detect malware attacks, utilizing SHAP for explainability. *Hernandes et al. (2021)* applied XAI methods like LIME and explainable boosting machines for phishing detection. *Karn et al. (2020)* advanced Cryptomining detection by combining SHAP, LIME, and an LSTM auto-encoding approach for interpretability. *Kalutharage et al. (2023)* proposed an XAI-based DDoS detection system, improving accuracy and attack certainty. *Morichetta, Casas & Mellia (2019)* applied LIME to analyze encrypted YouTube traffic, providing clear explanations of data clusters. An efficient deep neural network based models for detecting the intruders in networks has been presented along with merits of XAI in *Mane & Rao (2021)*. XAI algorithms like the contrastive explanations method (CEM), SHAP, LIME, ProtoDash and Boolean Decision Rules *via* Column Generation (BRCG) shed light on the "black box" of the NN, revealing which features trigger attack flags and to what extent. By applying these tools to the NSL-KDD dataset, the researchers demonstrate how XAI empowers security professionals to understand and trust the IDS's reasoning, boosting confidence in its defenses.

*Zebin, Rezvy & Luo (2022)* focused on analyzing encrypted traffic with the goal of accurately detecting DoH (DNS over HTTPS) attacks. A balanced stacked RF classifier was proposed as an effective solution for identifying such attacks. The research prioritized achieving high accuracy while ensuring transparency in the model's decision-making process. Performance improvements were largely attributed to the data split strategy and

the parallel development of sub-models, which led to a threefold reduction in training time. Moreover, an explainable artificial intelligence (XAI) model using SHAP was included to show each feature contributions to the classification decisions of the model. Building a bridge between performance and interpretability in ML, *Rabah, Le Grand & Pinheiro (2021)* proposed a novel framework. This framework boosts performance by addressing noisy, scattered, incomplete, and unbalanced data through the preprocessing phase. Notably, the Synthetic Minority Oversampling Technique (SMOTE) technique tackles data imbalance. To enhance interpretability, the framework leverages LIME for local explanations and permutation feature importance for global insights. Trust is further built by employing XAI techniques to reveal the features influencing individual predictions. Most of the security breaches. Targeting the transaction are carried out by the account holder in Ethereum platform. Hence, the monitoring the activities of account holder may greatly prevents the possible security breaches.

Table 1 reports various AI-enabled models with and without integrating XAI, alongside other essential components such as resampling techniques and hyperparameter optimization (HPO), all of which influence the mitigation of malicious activities and the prevention of cyber-attacks on networks. This table also highlights significant works by *Farrugia, Ellul & Azzopardi (2020)*, and *Zhou, Yan & Zhang (2022)*, who used XAI and HPO tools to mitigate malicious activities in Ethereum-based applications. *Aziz et al. (2022)* alone applied an oversampling technique to address the data sample imbalance. The remaining works presented in Table 1 leverage the merits of XAI, with some also incorporating data sampling techniques and HPO tools, though not all studies include these components. According to the analysis in Table 1, few studies have focused on designing interpretable ML models for detecting suspicious activities on permissionless Ethereum platform. Additionally, Table 1 presents details on the number of samples considered in the studies, the features used, and the different machine learning models employed. Upon analyzing the AI-enabled methods reported in various studies, the key takeaways are as follows: A comprehensive AI model should integrate essential components, such as feature interpretability tools, data resampling methods, and feature engineering techniques. Hence, this study addresses this gap by applying Explainable AI tools, oversampling technique and a simple hyperparameter optimization technique for designing the ensemble stacking model, balancing both performance and transparency, and setting a new direction for fraud detection research in blockchain security.

## MATERIALS AND METHODS

The creation of a cyber-attack detection model requires a systematic approach to guarantee its efficacy and dependability. This section outlines the process involved in building the proposed model. The architecture view as mentioned in Fig. 1, outlines the proposed XAI-enabled ensemble stacking-based cyber-attack detection framework intended to find security breaches on the Ethereum platform. ML models hyper-parameters are improvised through Bayesian optimization after the pre-processing. The ensemble stacking model is built upon leveraging the merits of RF, XGBoost, and NN. Since the ML models are worked on using a black box approach, to interpret their decisions in order to influence the

**Table 1 Technical specifications of cutting-edge methods.**

| Ref. | Blockchain used or not | Features | Samples count | XAI/RT | HPO | Attack | ML | XAI | RT |
|---|---|---|---|---|---|---|---|---|---|
| *Kumar et al. (2020)* | ☑ | 44 | 5,450 | – | – | Malicious accounts | LR, SVM, RF, AdaBoost, stacking | ☒ | ☒ |
| *Farrugia, Ellul & Azzopardi (2020)* | ☑ | 42 | 4,681 | SHAP | Grid search cv | Malicious account | XGBoost | ☑ | ☒ |
| *Kalutharage et al. (2023)* | ☒ | 14 | 66,793 | SHAP | – | DDoS | Autoencoder | ☑ | ☒ |
| *Zhou, Yan & Zhang (2022)* | ☑ | 42 | 4,681 | SHAP | Catboost hyperparameter optimization | Malicious account | CatBoost | ☑ | ☒ |
| *Aziz et al. (2022)* | ☑ | 17 | 9,841 | SMOTE | – | Malicious account | LGBM | ☒ | ☑ |
| *Ibrahim, Elian & Ababneh (2021)* | ☑ | 42 | 7,809 | – | – | Malicious account | KNN | ☒ | ☒ |
| *Morichetta, Casas & Mellia (2019)* | ☒ | 477 | 10,654 YouTube videos | LIME | – | Network traffic | Agglomerative_Wardizes, Agglomerative_Single, K-Means BIRCH | ☑ | ☒ |
| *Mane & Rao (2021)* | ☒ | 122 | 125,972 | SHAP, LIME, CEM, ProtoDash and BRCG | – | DoS, Probe, R2L, U2R | DNN | ☑ | ☒ |
| *Zebin, Rezvy & Luo (2022)* | ☒ | 82 | 2.1 million | SHAP/ SMOTE | – | DoH | RF | ☑ | ☑ |
| *Rabah, Le Grand & Pinheiro (2021)* | ☒ | 115 | 652,100 | LIME/ SMOTE | Default parameters | Mirai and Bashlite Malware | DT, KNN, SVM, MLP, RF, and ET | ☑ | ☑ |
| *Wang et al. (2020)* | ☒ | 41 | 125,972 | SHAP & LIME/ SMOTE | – | DoS, Probe, R2L, U2R | One-*vs*-All classifier/ multiclass classifier | ☑ | ☒ |
| *Wali & Khan (2023)* | ☒ | 78 | 16,000,000 | SHAP | Randomized search | DoS HULK, DoS SlowHTTP, SSH Brute Force, DoS HOIC, FTP Brute Force, DoS LOIC-UDP | RF | ☑ | ☒ |
| *Sarhan, Layeghy & Portmann (2022)* | ☒ | 170 | 109,700,000 | SHAP | – | Network attacks | DFF, RF | ☑ | ☒ |
| *Le et al. (2022)* | ☒ | 169 | 2,889,295 | SHAP | Default parameters | DoS, Web-based | Ensemble tree | ☑ | ☒ |
| *Alani (2022)* | ☒ | 35 | 1,503,895 | SHAP | – | Botnet attack | XGB | ☑ | ☒ |

(Continued)

| Ref. | Blockchain used or not | Features | Samples count | XAI/RT | HPO | Attack | ML | XAI | RT |
|---|---|---|---|---|---|---|---|---|---|
| Proposed | ☑ | 17 | 12,411 | SHAP & LIME & ELI5 | Bayesian optimization | Malicious accounts | Ensemble stacking classifier | ☑ | ☑ |

**Note:**
XAI = eXplainable Artificial Intelligence; RT = Resampling Technique.

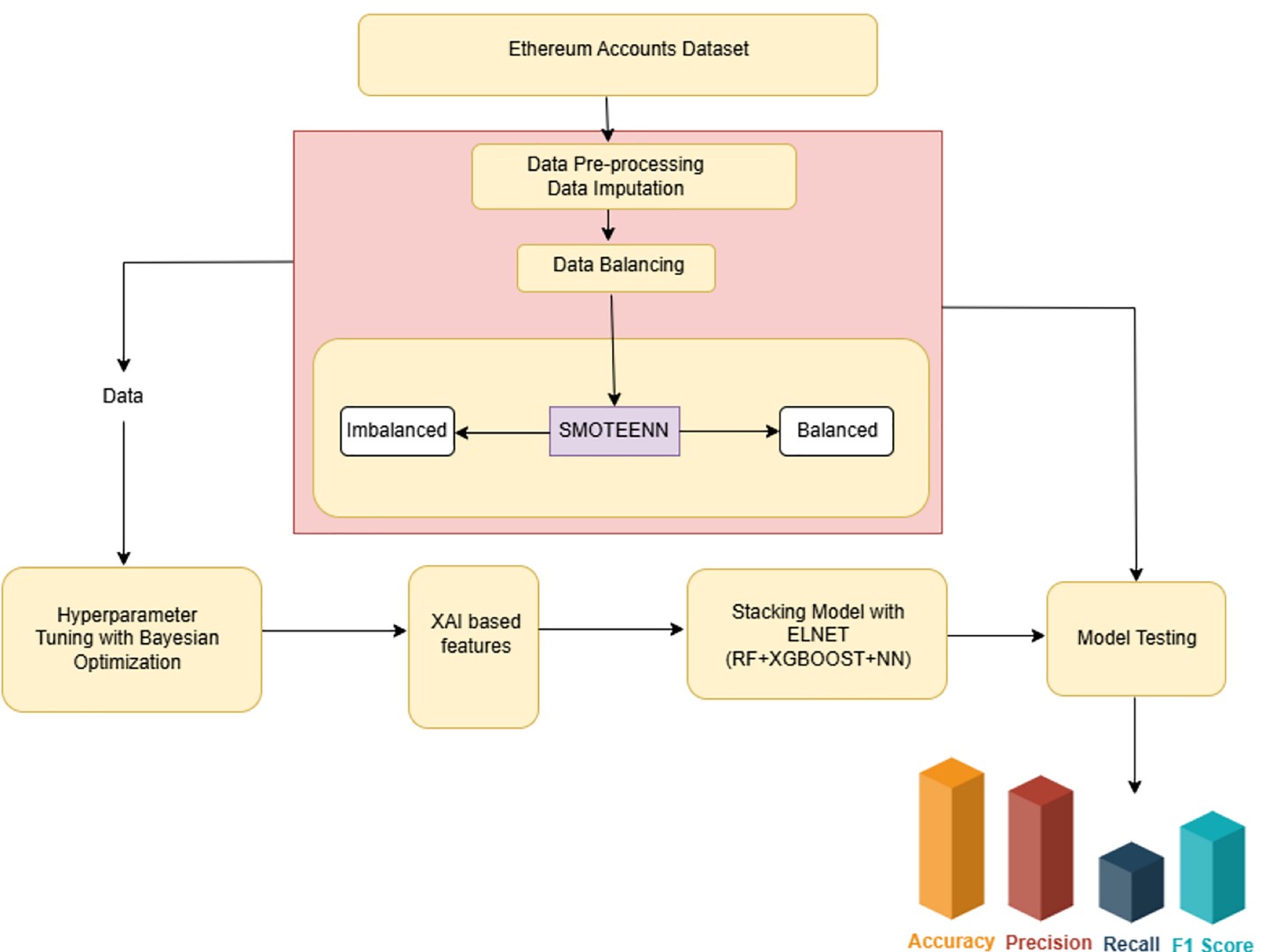

**Figure 1** Architectural view of the candidate framework.

**Table 2 Dataset description.**

| S.no | Feature name | Description |
|---|---|---|
| 1 | FLAG | Indicates whether the transaction is fraud or not |
| 2 | Avg min between sent tnx_F1 | Average time between sent transactions for the account in minutes |
| 3 | Avg min between received tnx_F2 | Average time between received transactions for the account in minutes |
| 4 | Time Diff between first and last(Mins)_F3 | Time difference between the first and last transaction in minutes |
| 5 | Sent tnx_F4 | Total number of sent normal transactions |
| 6 | Received tnx_F5 | Total number of received normal transactions |
| 7 | Number of Created Contracts_F6 | Total Number of created contract transactions |
| 8 | Max Value Received_F7 | Maximum value in Ether received |
| 9 | Avg Value Received_F8 | Average value in Ether received |
| 10 | Avg Val Sent_F9 | Average value of Ether sent |
| 11 | Total Ether Sent_F10 | Total Ether sent from the account address |
| 12 | Total Ether Balance_F11 | Total Ether Balance after all transactions |
| 13 | ERC20 Total Ether Received_F12 | Total ERC20 token received transactions in Ether |
| 14 | ERC20 Total Ether Sent_F13 | Total ERC20 token sent transactions in Ether |
| 15 | ERC20 Total Ether Sent Contract_F14 | Total ERC20 token transfer to other contracts in Ether |
| 16 | ERC20 Uniq Sent Addr_F15 | Number of ERC20 token transactions sent to unique account addresses |
| 17 | ERC20 Uniq Rec Token Name_F16 | Number of Unique ERC20 tokens received |

prediction, the XAI tool has been included. And the feature importance derived through diverse XAI enabled stacking model has been used to train the proposed model.

## Dataset description and preprocessing

The dataset used for experimentation, as documented from *Aliyev (2020)*, comprises 9,841 entries encompassing both benign and malign Ethereum transactions. Initially, dataset consists of 52 fields, the preprocessing steps removed irrelevant columns. And the resultant columns descriptions can be found in Table 2.

The dataset, featuring 17 selected columns relevant to anomaly detection (*Chithanuru, 2023*), includes a binary target variable, *Flag*, with 0 indicating non-illicit accounts and 1 indicating illicit accounts. It was divided in an 80:20 ratio between training and testing sets. Class distribution analysis revealed significant imbalance, with legitimate transactions dominating. To address this, SMOTEENN (Synthetic Minority Oversampling Technique-Edited Nearest Neighbors) was applied to balance the dataset (*Isangediok & Gajamannage, 2022*). The method produces synthetic examples for the underrepresented class, improving both accuracy and generalizability. After resampling, the dataset comprised 6,055 non-illicit and 6,356 illicit samples.

## Exploratory data analysis and scaling

To prepare the data for modeling, we conducted an exploratory data analysis (EDA) to understand the sample distribution and feature characteristics. EDA revealed that scaling the features was necessary to ensure uniformity across data ranges, which enhances the

model's ability to learn effectively. Based on our findings, min-max normalization was chosen for scaling. This normalization technique rescales feature values to a specific range, typically [0, 1], which helps prevent certain features from disproportionately influencing the model. The candidate dataset, with its comprehensive set of features and adjustments for class imbalance, provides a robust foundation for detecting fraudulent Ethereum transactions.

Features that captures essential transactional and behavioral data that support detecting suspicious activities in network or blockchain contexts is highly needed. For instance, in the resultant 17 columns, the "Time Difference between First and Last Transactions" helps identify bursts of activity often associated with DoS or botnet attacks. Tracking "Unique ERC20 Token Names Received" and "Total Ether Received" can reveal unusual token interactions or abnormal transaction volume, which may suggest phishing or laundering attempts. Monitoring "Average Time between Received Transactions" and "Average Transaction Value Received" can highlight irregular transaction patterns, a common sign of account compromise. Additionally, observing "Total Ether Balance" and "Total Ether Sent" allows for spotting unexpected inflows or outflows of funds, often indicating fraud or malicious behavior. These metrics together offer a detailed profile of normal *vs*. potentially harmful transactional behaviors, enhancing accuracy in attack detection.

## Hyperparameter tuning with Bayesian optimization

The ML model's behavior and performance are significantly influenced by the external configuration settings referred to as hyperparameters. Bayesian optimization is a technique used in hyperparameter tuning for ML models. This iterative approach seeks to identify the best set of hyperparameters by effectively balancing the exploration and exploitation of the hyperparameter space (*Demircioğlu & Bakır, 2023*; *Paudel, Montoya & Mandal, 2023*). Exploration involves searching the hyperparameter space widely to discover new regions that may contain better-performing hyperparameter configurations. Exploitation involves focusing on areas of the hyperparameter space that are currently believed to be more likely to contain optimal or high-performing hyperparameter configurations (*Demir & Sahin, 2023*; *Albahli, 2023*). The process of selecting the hyperparameters using the Bayesian optimization as follows and shown in the below Fig. 2.

1) **Initial sampling:** Bayesian optimization starts with an initial set of hyperparameter configurations, often chosen randomly or using a simple heuristic.

2) **Modeling the objective function:** Bayesian optimization leverages a surrogate model, often a Gaussian process, to represent the objective function—such as validation accuracy or loss—based on hyperparameters. This surrogate model offers predictions of the objective function and its associated uncertainty.

   The Gaussian process (GP) regression model is commonly used as a surrogate model in Bayesian optimization. Given a set of observed data points $(x_i, y_i)$ where $x_i$ are hyperparameter configurations and $y_i$ are corresponding objective function values, the

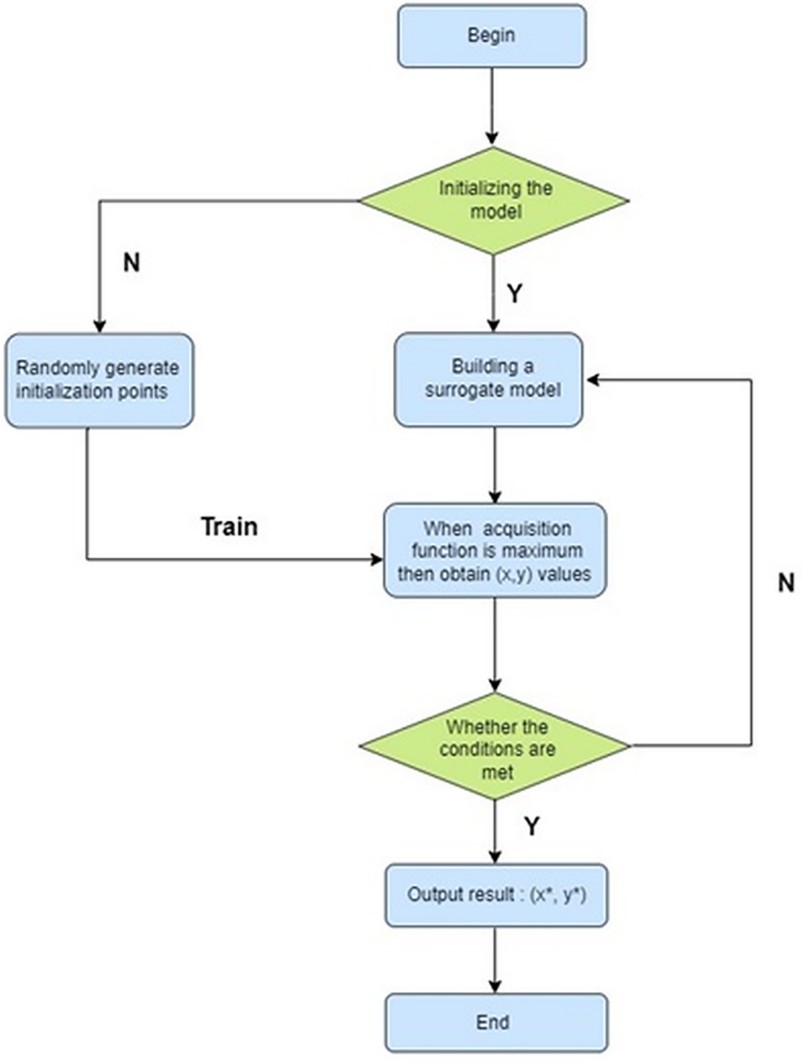

**Figure 2 Flowchart of Bayesian optimization.**

GP model predicts the objective function *f(x)* at point *x* newly as a Gaussian distribution:

$$F(x) = \mathrm{N}(\mu(x),\ \sigma^2(x)) \tag{1}$$

where:

$\mu(x)$ is the GP mean function, representing predicted objective function value at *x*.
$\sigma^2(x)$ is variance function of the GP, representing the uncertainty or confidence in the prediction at *x*.

3) **Acquisition function:** Using the surrogate model, an acquisition function such as Expected Improvement or Upper Confidence Bound is employed to select the next set of hyperparameters for evaluation. This function manages the trade-off between

exploring new configurations and exploiting the most promising ones. Commonly used acquisition functions include:

(a). Expected improvement (EI):

$$EI(x) = E[\max(0, f_{min} - f(x))] = (\mu(x) - f_{min})\Phi(z) + \sigma(x)\phi(z) \qquad (2)$$

where, $f_{min}$ is the minimum observed objective function value. $\Phi(z)$ is the cumulative distribution function of the standard normal distribution. $\phi(z)$ is the probability density function of the standard normal distribution. $z = \dfrac{\mu(x) - f_{min}}{\sigma(x)}$ is the standardization of the predicted improvement.

(b). Upper confidence bound (UCB):

$$UCB(x) = \mu(x) + \beta\sigma(x) \qquad (3)$$

where, $\beta$ is a tunable parameter that balances exploration (higher values of $\beta$) and exploitation (lower values of $\beta$).

(c). Probability of improvement (PI):

$$PI(x) = \Phi\left(\frac{\mu(x) - f_{min} - \xi}{\sigma(x)}\right) \qquad (4)$$

where, $\xi$ is a parameter that controls the trade-off between exploitation and exploration.

4) **Evaluation:** The selected hyperparameter configuration is evaluated using the actual objective function (*e.g.*, training on a subset of data and validating on a separate validation set).

5) **Update surrogate model:** The surrogate model is refined using the newly acquired data point, which includes the hyperparameter configuration and its corresponding objective function value.

6) **Iterate:** Steps 3–5 are continuously executed until a predefined convergence criterion is satisfied, such as reaching a specific number of iterations.

## Machine learning models

This section briefs the designing process involved in building the ensemble stacking models. Models like RF, XGBoost and NN are considered as a base models for the presented experimentation.

## Random forest

Random Forest is a highly effective ensemble machine learning method commonly used for both classification and regression tasks. In contrast to a single decision tree, which can easily overfit and display high variance, RF creates multiple decision trees during the training process and combines their outcomes. This approach enhances both accuracy and generalization.

### eXtreme gradient boosting

XGBoost stands out as a top ML algorithm, known for its speed and ability to tackle various problems. It takes a unique approach to gradient boosting, making it efficient and accurate. XGBoost shines in its ability to handle different data types and uncover complex patterns within them. It achieves this by combining multiple, simple models (often decision trees) and progressively refining its predictions.

### Neural network

Neural networks, inspired by the structure and functionality of the human brain, are a powerful type of machine learning model frequently applied to tasks such as classification, regression, and clustering. They consist of interconnected nodes organized into layers, including an input layer, one or more hidden layers, and an output layer. Each node processes input data by applying weights, summing the weighted inputs, and passing the result through an activation function to produce an output. The weights between nodes determine the influence of one node on another, while activation functions, such as ReLU (rectified linear unit), sigmoid, tanh (hyperbolic tangent), and softmax, introduce non-linearity, allowing the network to capture complex data patterns. Neural networks are trained using optimization techniques like gradient descent and its variants (*e.g.*, Adam and RMSprop), which iteratively update the weights to minimize the error between predicted and actual values, as defined by a loss function. Backpropagation is a critical algorithm used during training to compute gradients efficiently and adjust the weights accordingly.

### Ensemble stacking model

Stacking stands out as a potent ensemble learning method that amalgamates the predictions generated by several individual models, culminating in a final prediction that is not only more resilient but also more accurate (*Chen et al., 2020a*; *Nayyer et al., 2023*). Training ensemble model includes two levels. Firstly, base learners are the individual models trained on the original data. In this case, we consider three different models XGBoost, NN, and RF. The main element of ensemble stacking is meta-learner. Upon the completion of training for the base learners, their predictions are employed as input to facilitate the training of a meta-learner. The meta-learner is trained to integrate predictions from the base learners to produce a final output. Elastic Net (ELNet) is used in this with meta-learner. ELNet regularization is a technique utilized for both regression and classification that integrates L1 and L2 regularization methods. The combination of these two techniques often results in improved performance compared to using each one individually. ELNet regularization helps minimize the variance of the final model, enhancing its robustness to noise and outliers. Additionally, ELNet models provide coefficients that reflect the relative importance of each feature, making them more interpretable compared to other ensemble models such as RF.

XAI techniques is an efficient as well as reliable tool in order to interpret the decisions made by ML models. In particular, feature importance derived through the XAI tool offers better insights to detect the anomalous activities (*Capuano et al., 2022*). For the candidate

framework, merits of the XAI tools, SHAP, Lime, and ELI5 have been included to enhance the interpretability of the presented ensemble stacking model. SHAP's unique approach guarantees fairness in attributing responsibility for the model's fraud detection to individual features (*Mane & Rao, 2021*; *Zebin, Rezvy & Luo, 2022*). LIME fits an interpretable model around a given instance to produce locally faithful explanations for model predictions. Work presented by *Morichetta, Casas & Mellia (2019)*, *Mane & Rao (2021)*, *Rabah, Le Grand & Pinheiro (2021)* leverages the merits of LIME to generate local explanations for its intended tasks. ELI5 is model-agnostic, allowing it to be utilized with a variety of machine learning models. Utilizing ELI5 allows for the visualization and interpretation of the significance of various features within a dataset, contributing to the identification of the anomalous entity. Incorporating XAI tools—SHAP, LIME, and ELI5—into the candidate model allowed for a deeper analysis of feature importance and model transparency. SHAP values provided a global perspective on feature impacts across the dataset, showing how transaction frequency and account existence contribute to fraud detection. LIME further enabled local explanations by fitting interpretable models around individual predictions, illustrating how features like Ether volume and timing of transactions affect specific accounts. ELI5 helps to visualize and validate the significance of various features within the dataset, thereby enhancing the model's interpretability and accountability. In contrast, the presented experiment integrates the features produced by all three libraries. The XAI libraries like SHAP, LIME, and ELI5 consistently identified features F2, F3, F4, F8, F12, and F16 as significant. Additionally, SHAP highlighted features F5, F7, F10, and F11 were included due to their valuable insights into both global and local feature importance, as well as their ability to effectively handle feature interactions as described in Table 2. Hence, the demonstrated influential features upon the Feature importance derived through the various XAI tools, SHAP, LIME, and ELI5, is displayed in Figs. 3A–3C.

Introducing an ElasticNet meta-learner within the ensemble stacking framework effectively balances complexity, especially with high-dimensional data, by reducing computational overhead. Bayesian optimization further enhances this framework by fine-tuning hyperparameters, achieving an optimal trade-off between exploration and exploitation. XAI tools validate the contributions of base models, mitigating overfitting while ensuring computational efficiency. This integration enables the meta-learner to combine predictions efficiently, capitalizing on the strengths of each base model and mitigating their weaknesses, ultimately producing a robust and efficient model. Key features influencing fraudulent transaction detection include ERC20 total ether received and ERC20 unique received token names, with higher values strongly indicating fraud. Features like time difference between first and last transaction and average minutes between received transactions also significantly impact predictions, with context-dependent effects. Metrics such as average value received and total ether sent further aid in identifying suspicious activities. The ranking of features highlights ERC20 unique received token names as the most influential, followed by time difference between first and last transaction, emphasizing their critical roles in determining transaction legitimacy. The

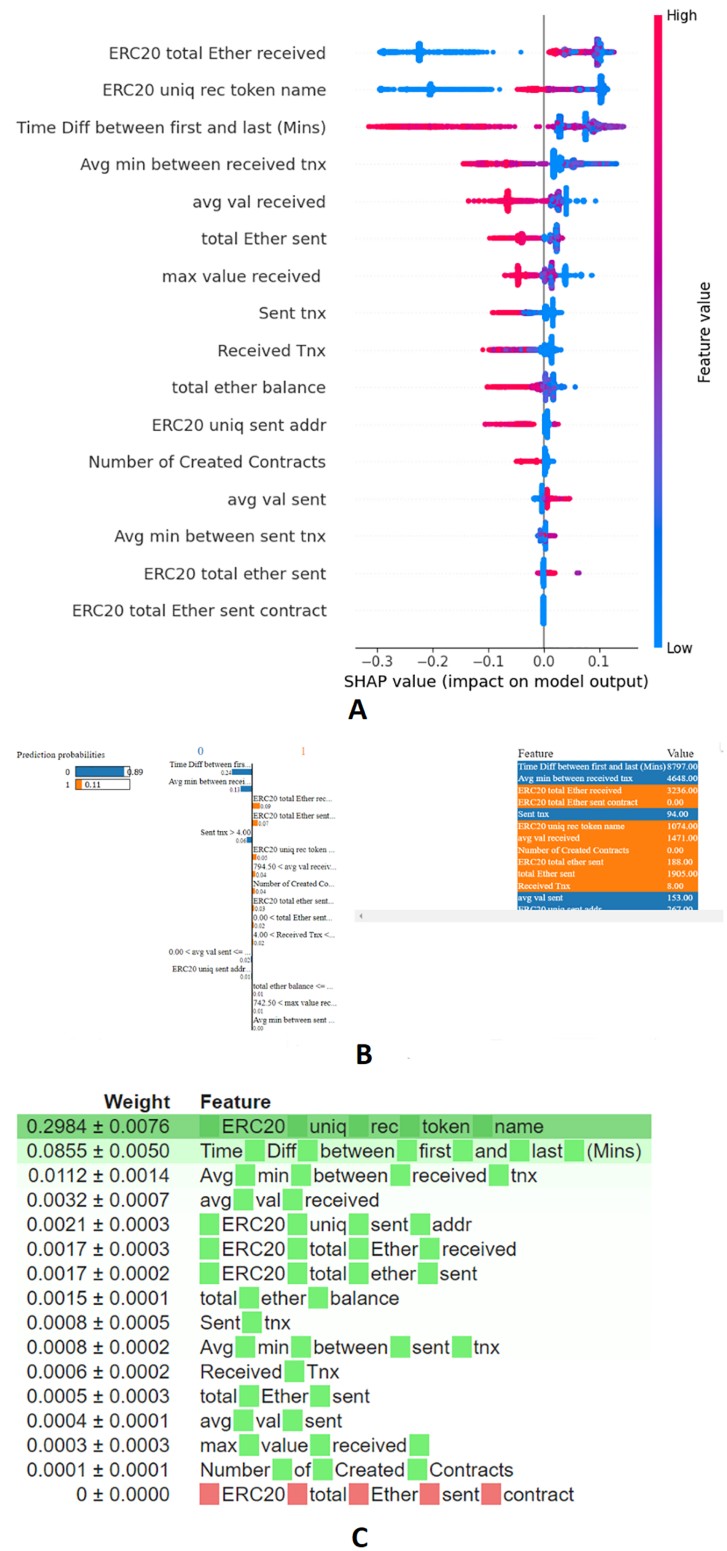

**Figure 3  Feature importance through XAI-enabled stacking model: (A) SHAP, (B) LIME, (C) ELI5.**

**Table 3 Hyperparameters of machine learning and XAI.**

| Hyperparameters | Range | RF | XGBoost | NN | Ensemble stacking classifier |
|---|---|---|---|---|---|
| n_estimators | [80, 150] | 123 | 131 | – | – |
| max_depth | [3, 10] | 10 | 10 | | |
| max_features | [0.1, 1.0] | 0.3139 | – | – | – |
| min_samples_leaf | [2, 20] | 1.7307 | – | – | – |
| min_samples_split | [1, 10] | 14.0528 | – | – | – |
| colsample_bytree | [0.8, 1.0] | – | 0.8 | – | – |
| learning_rate | [0.01, 1.0] | – | 1.0 | – | – |
| min_child_weight | [1, 10] | – | 1 | – | – |
| subsample | [0.8, 1.0] | – | 0.8 | – | – |
| activation | [relu, sigmoid] | – | – | Relu | – |
| batch_size | [32, 64, 128] | – | – | 32 | – |
| epochs | [10, 20, 30] | – | – | 30 | – |
| num_hidden_layers | [1, 2, 3] | – | – | 3 | – |
| num_units | [32,64, 128] | – | – | 64 | – |
| optimizer | [adam, sgd] | – | – | adam | – |
| l1_ratio | [0, 1] | – | – | – | [0.5] |
| alphas | [0.0001, 10.0] | – | – | – | [0.1] |
| Cv | 5 | 5 | 5 | 5 | 5 |
| max_iter | [100, 5,000] | – | – | – | 1,000 |
| Tol | [1e−5, 1e−3] | – | – | – | 0.0001 |
| fit_intercept | [True, False] | – | – | – | True |
| selection | Cyclic, random | – | – | – | Cyclic |
| n_jobs | [1, −1] | – | – | – | −1 |

**XAI control parameters**

| | Range | SHAP | LIME | XAI | – |
|---|---|---|---|---|---|
| Method | – | TreeExplainer | LimeTabularExplainer | explain_weights | – |
| model_output | [raw, probability, predict_proba, predict] | Raw | – | – | – |
| feature_perturbation | [random, interventional, none] | Interventional | – | – | – |
| kernel_width | [0.1, 1.0] | – | 0.4 | – | – |
| feature_selection | [auto, none] | – | auto | – | – |
| discretize_continuous | [true, false] | – | True | – | – |
| sample_around_instance | [true, false] | – | False | – | – |
| importance_type | [weight, shap, permutation, gain] | – | – | gain | – |

hyperparameters of ML, ensemble stacking classifier, and XAI control parameters are depicted in Table 3.

## RESULTS AND DISCUSSION

The effectiveness of the proposed ensemble learning model was assessed by benchmarking it against several well-known models using quantitative performance metrics, including precision (Pr), recall (Re), F1-score (F1), and accuracy (Ac). These metrics were derived

**Table 4 Tested results.**

| Model | Sample | Recall | Precision | F1-score | Accuracy |
|---|---|---|---|---|---|
| RF | Actual | 0.9265 | 0.9606 | 0.9433 | 0.9761 |
| RF | SMOTE + ENN | 0.9951 | 0.9935 | 0.9943 | 0.9943 |
| XGB | Actual | 0.9454 | 0.9637 | 0.9545 | 0.9807 |
| XGB | SMOTE + ENN | 0.9967 | 0.9959 | 0.9963 | 0.9963 |
| NN | Actual | 0.9250 | 0.8777 | 0.9007 | 0.9558 |
| NN | SMOTE + ENN | 0.9412 | 0.9421 | 0.9411 | 0.9412 |
| Proposed | Actual | 0.9532 | 0.9465 | 0.9432 | 0.9843 |
| Proposed | SMOTE + ENN | 0.9963 | 0.9962 | 0.9952 | 0.9960 |

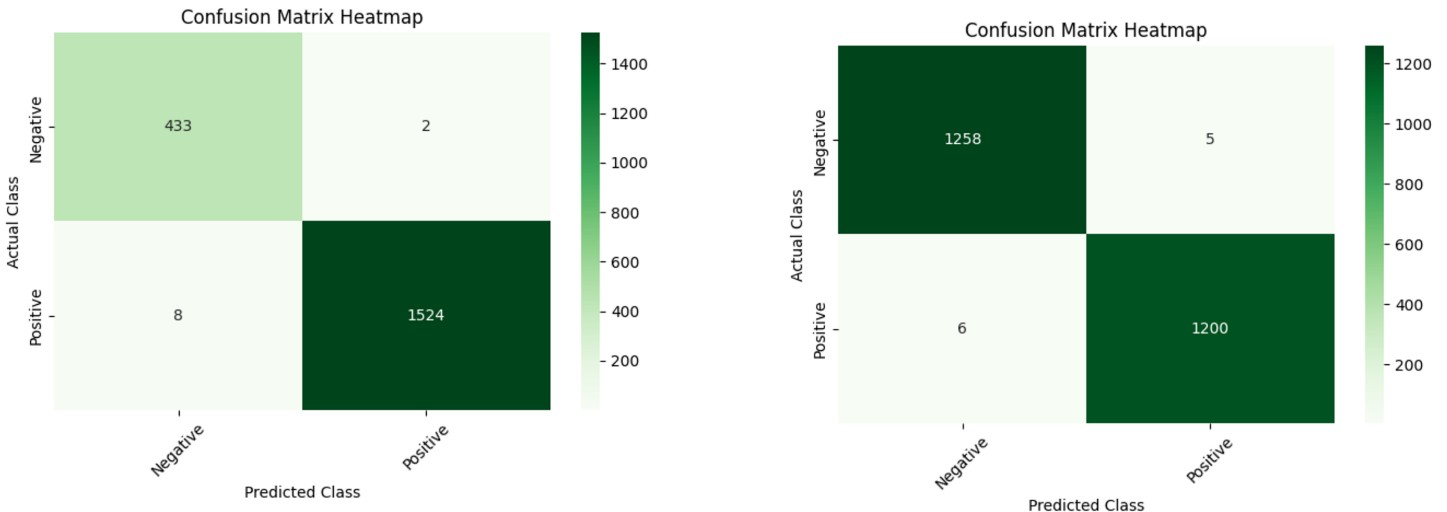

**Figure 4 Ensemble stacking output before (A) and after (B) applying resampling techniques.**

from the counts of true positives (Tp), true negatives (Tn), false positives (Fp), and false negatives (Fn). The experiments were conducted using the dataset (*Aliyev, 2020*; *Chithanuru, 2023*) and implemented with the Keras and TensorFlow libraries in Python 3.3.4. The experimental setup consisted of a Windows 10 operating system, an Intel Core i5-8250U processor, and 32 GB of RAM.

Table 4 shows the performance of ML models (RF, XGB, NN, and Stacking) for detecting fraudulent accounts on Ethereum, comparing results before and after applying SMOTEENN. The models exhibit improved performance after resampling. A comparison of ML models with the ensemble stacking model shows an accuracy of 99.6%. The improved performance is attributed to efficient feature selection using XAI tools. Features like number of created contracts and ERC20 unique sent address help detect spam contracts, phishing, and money laundering. Transaction behavior features such as time difference between first and last transaction and average value received highlight

**Peer**J Computer Science

**Table 5 Computational cost.**

| | Training time (in sec) | Memory used during training (MB) | Testing time (in sec) | Average mean prediction | Average standard deviation |
|---|---|---|---|---|---|
| **For 17 features** | | | | | |
| RF | 3.72 | 1.59 | 0.04 | 0.4981 | 0.0073 |
| XGB | 1.93 | 1.53 | 0.01 | 0.4979 | 0.0086 |
| NN | 4.85 | 12.94 | 0.41 | 0.3523 | 0.2632 |
| Ensemble stacking | 0.21 | 0.0 | 0.21 | 0.5073 | 0.0168 |
| **For 10 features** | | | | | |
| RF | 2.16 | 1.29 | 0.03 | 0.4984 | 0.0079 |
| XGB | 1.36 | 1.29 | 0.02 | 0.4982 | 0.0100 |
| NN | 4.66 | 11.76 | 0.36 | 0.3121 | 0.2378 |
| Ensemble stacking | 0.20 | 0.00 | 0.21 | 0.5113 | 0.0057 |

suspicious activity patterns indicative of fraud. Figure 4 also provides snapshots of the confusion matrices, which is highly appreciated in the examination of the malicious Ethereum accounts.

## Computational cost analysis

From a computational cost and uncertainty perspective, RF and the ensemble stacking model stand out as the most effective choices, with the ensemble model being particularly reliable as shown in Table 5. RF is highly efficient, with low training and testing times, minimal memory usage (1.29–1.59 MB), and stable predictions characterized by a small standard deviation in mean predictions, indicating low uncertainty. It performs consistently well across both 17 and 10 features, showcasing its adaptability and scalability. The ensemble stacking model, on the other hand, excels by leveraging the strengths of multiple models, achieving robust predictions with minimal computational overhead. Its training and testing times are fast, and memory usage is negligible, while the low standard deviation of predictions demonstrates its ability to minimize uncertainty, making it the most balanced and reliable approach. In contrast, the NN requires significantly more resources, with high memory consumption (up to 12 MB) and longer training times, and it exhibits increased uncertainty, particularly when the number of features is reduced, as shown by the sharp rise in standard deviation. This suggests that the NN might be overfitting or struggling to generalize with fewer features, requiring further optimization of its architecture or hyperparameters. While NN has potential for complex patterns, its computational cost and uncertainty make it less suitable for resource-constrained environments or applications demanding consistent predictions. Therefore, for scenarios where computational efficiency and prediction stability are critical, RF and the ensemble stacking model are superior choices, with the latter being the most robust and

**Table 6 Benchmarking results against cutting-edge techniques.**

| Reference | Models | Number of features | Accuracy |
|---|---|---|---|
| *Kumar et al. (2020)* | XGB | 10 | 96.8 |
| *Farrugia, Ellul & Azzopardi (2020)* | XGB | 42 | 96.3 |
| *Zhou, Yan & Zhang (2022)* | CatBoost | 43 | 94.0 |
| *Aziz et al. (2022)* | LGBM | 43 | 99.0 |
| *Ibrahim, Elian & Ababneh (2021)* | KNN | 42 | 98.7 |
| *Ibrahim, Elian & Ababneh (2021)* | J48 | 6 | 97.9 |
| Proposed | XAI-ensemble stacking classifier | 10 | 99.6 |

cost-effective due to its ability to aggregate and mitigate individual model uncertainties while maintaining excellent performance.

## Performance comparison with state-of-the-art methods

This subsection assesses the accuracy of the proposed ensemble stacking model by benchmarking it against leading-edge techniques. The findings shown in Table 6, once again confirmed that the presented XAI enabled Ethereum malicious account detection is superior than that of it counter-part techniques.

## Limitations and future directions

The results of the presented Ethereum-based fraudulent activity detection model are superior in terms of various quantitative metrics. The dataset used for experimentation contains transaction details from the years 2017 to 2019. Recently, the landscape of dynamic threats has been increasing day by day. Therefore, to make reliable predictions in the coming years, continuous learning is necessary, along with the inclusion of new attack sample vectors.

Future research could compile a new dataset by including various attack features collected after 2019. The inclusion of other possible features that could better describe the attacks would also be investigated. Designing deep learning techniques requires more samples; therefore, an effort will be made to collect a larger number of samples.

## CONCLUSION

This research focuses on proactively examining user behavior on the Ethereum platform, a permissionless blockchain, to reduce potential cyber threats like eclipse attacks, phishing, sybil attacks, Ponzi schemes, and DDoS attacks. The proposed framework tackles overfitting and underfitting issues using suitable data sampling techniques. Hyperparameters are critical variables that greatly affect the performance of machine learning models. Therefore, we utilize Bayesian optimization to optimize these hyperparameters. Furthermore, XAI tools are employed to identify key features that boost the reliability of predictions made by the ensemble stacking model. This model, enhanced by XAI-derived features, is trained and validated on an Ethereum dataset. The findings show that the proposed framework achieves an impressive 99.6% accuracy, surpassing

other evaluated frameworks. To gain a deeper understanding of anomalous account behavior on the Ethereum platform, future research will include transaction data from 2021, 2022, and 2023. This extended dataset will enable us to explore the potential of deep learning techniques for even more effective anomaly detection.

### Funding
This work was supported by the Vellore Institute of Technology. The funders had no role in study design, data collection and analysis, decision to publish, or preparation of the manuscript.

### Grant Disclosures
The following grant information was disclosed by the authors:
Vellore Institute of Technology.

### Competing Interests
The authors declare that they have no competing interests.

### Author Contributions
- Vasavi Chithanuru conceived and designed the experiments, performed the experiments, analyzed the data, performed the computation work, prepared figures and/or tables, and approved the final draft.
- Mangayarkarasi Ramaiah analyzed the data, authored or reviewed drafts of the article, and approved the final draft.

### Data Availability
The data is available at Kaggle: https://www.kaggle.com/datasets/vasavichithanuru/ethereum-fraud-transactions.

The dataset is available at GitHub and Zenodo:

- https://github.com/vasavichithanuru/XAI-Enabled-Ensemble-Stacking.

- vasavichithanuru. (2024). vasavichithanuru/XAI-Enabled-Ensemble-Stacking: 3374d4d (3374d4d). Zenodo. https://doi.org/10.5281/zenodo.14280345.

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
