# Peer review of "Proactive detection of anomalous behavior in Ethereum accounts using XAI-enabled ensemble stacking with Bayesian optimization"

_PeerJ Computer Science, doi:10.7717/peerj-cs.2630_

## Round 0.1 · original submission · Major Revisions

Dear authors,

Thank you for submitting your article. Feedback from the reviewers is now available. Your article has not been recommended for publication in its current form. However, we do encourage you to address the concerns and criticisms of the reviewers and resubmit your article once you have updated it accordingly. Before submitting the revised paper, following should also be addressed:

1. . Equations should be used with correct equation number. Numbering is started from (4) in the manuscript. Furthermore, many of the equations are part of the related sentences. Attention is needed for correct sentence formation.
2. All of the values for the parameters of all algorithms should be given.
3. Advantages, disadvantages and limitations of the proposed method should be provided.
4. Figures should be polished. Many of them have low resolution.

Best wishes,


**Language Note:** The Academic Editor has identified that the English language must be improved. PeerJ can provide language editing services - please contact us at [email protected] for pricing (be sure to provide your manuscript number and title). Alternatively, you should make your own arrangements to improve the language quality and provide details in your response letter. – PeerJ Staff

·

Basic reporting

Abstract is the main attractions for the readers to proceed to the remaining content of the article. However, there is no mention of Blockchain.

>Abstract should have the word Blockchain to understand reader on Ethereum platform,
> Keccak-256 algorithm citation should be added in Introduction.
> Kaggle data should contain data description not just the file. Reader may be interested to see the data description and may not grow interest if needed to download the file.
> GitHub codes should have comments to readers to understand better on the experiments.
> Paper reference links are showing to the local hard drive below. Should be modified to the actual URLs
file:///C|/.../PeerJ-research-manuscript-template.docx

Experimental design

Experimental Design can be further expanded
> Explain the data clean up process more detail
> Detail process of each models and comparisons with existing experiments explained in Literature Review.
> Benchmarking against Cutting-Edge Techniques should be more clearly addressed in Abstract.

Validity of the findings

> Results and Discussion can be further expanded.
> Research limitations and Future work should be addressed in conclusion.

Additional comments

Excellent work.

Reviewer 2 ·

Basic reporting

Basic reporting:

In title, the terms "usingXAI" should be presented as "using XAI".

In Abstract, DDoS term should be defined in order to help the readers.

In Abstract, there is a need to present the dataset details for the experimental evaluation. In addition, study implications should be presented.

In introduction, there is a lack of motivation. What are the limitations of the existing techniques to safeguard the network? How the proposed study intends to address those limitations?

In Major contributions, the first contribution is obvious for the research article. Thus, it is not required to be mentioned in the introduction.

The expansion of XGBoost should be presented in Abstract. Kindly remove the expansion of XGboost in literature review in line number 141.

There is a lack of knowledge gaps in literature review. The use of literature review is to provide the literature gaps and why we move forward to conduct this research.

In line number: 241, the authors mentioned the dataset description. It should provide the features of the dataset. For instance, what are the impressions of attack and how the proposed study use these features in explaining the attacks.

The use of meta-learner is appreciable. However, it will add substantial layer of complexity to the model. How the trade-off executed to maintain the optimal prediction.

The mathematical expressions are explored already in the literature. Thus, the novel expressions should be retained and other expressions should be removed.

The Algorithm is useful. However, i did not notice the source code for the proposed model. It is advisable to include the source code for further exploration.

In results and discussion, Eqn. 4 - 7 should be removed. It is already known.

In results and discussion, the uncertainty analysis should be presented. In addition, the computational costs with ablation study need to be included. It will be useful in the Ensemble leanring based studies to understand the performance of each model.

Experimental design

There is a lack of approach for computing the interpretable ability using LIME, ELI5 and SHAP values. For instance, how we know the specific feature is infuencing the model's ability.

Validity of the findings

A dedicated section should be introduced in order to present the study implications, limitations and future directions.

---

## Round 0.2 · accepted · Accept

Dear Authors,

One reviewer has accepted the manuscript. The invitation to review the revised manuscript was not responded to by one of the previous reviewers. I have assessed the revision myself and, in my view, your paper is now sufficiently improved following the last revision. It is therefore ready for publication. By the way, please pay special attention for correct sentence formation. Many of the equations are part of the related sentences. Please correct them and upload minor corrections in production phase.

Best wishes,

Reviewer 2 ·

Basic reporting

The authors have addressed my concerns and improved the manuscript's standard. I thank them for their efforts. The current form of the manuscript may be acceptable for the publication.

Experimental design

The authors have addressed my concerns and improved the manuscript's standard. I thank them for their efforts. The current form of the manuscript may be acceptable for the publication.

Validity of the findings

The authors have addressed my concerns and improved the manuscript's standard. I thank them for their efforts. The current form of the manuscript may be acceptable for the publication.

Additional comments

The authors have addressed my concerns and improved the manuscript's standard. I thank them for their efforts. The current form of the manuscript may be acceptable for the publication.